# Reconstructing Template-Memorized Images from Natural Prompts

**Sol Yarkoni** [1]  **Mahmood Sharif** [2]  **Roi Livni** [1]

## Abstract

Recent advances in generative models, such as diffusion models, have raised concerns related to privacy, copyright infringement, and data curation. Prior work has shown that training data can be reconstructed from such models, but existing attacks typically rely on substantial computational resources, access to the training set, or carefully engineered prompts.

In this work, we present a low-resource reconstruction attack that operates through seemingly benign prompts and requires little to no access to the training data. Our attack targets *template-memorized images* (TMI), where recurring layouts and visual structures are memorized during training. We show that such memorization manifests under potentially realistic usage. This raises a possibility of unintentional reconstruction by naïve users that do not carry explicit adversarial intent. For example, we observe that a simple prompt such as "blue Unisex T-Shirt" can reproduce visual content depicting a real individual. Beyond extraction, we observe novel phenomena occurring in TMI (e.g., interpolation), raising questions about the novelty of generated content and the effectiveness of established methods for detecting memorized content.

Our code is available at https://github.com/TheSolY/lr-tmi.

## 1. Introduction

With the growing adoption of generative and foundation models, large-scale public, and occasionally private, data are increasingly used for training. This practice raises serious concerns around privacy, copyright, and consent, as

[1]School of Electrical & Computer Engineering, Tel Aviv University [2]School of Computer Science & AI, Tel Aviv University. Correspondence to: Sol Yarkoni <solyarkoni@mail.tau.ac.il>.

*Proceedings of the 43rd International Conference on Machine Learning*, Seoul, South Korea. PMLR 306, 2026. Copyright 2026 by the author(s).

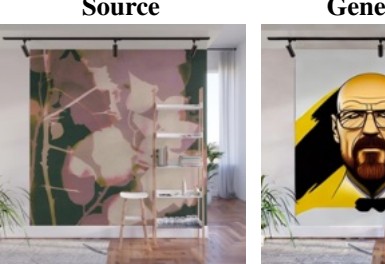

**Source**    **Generated**

*Figure 1.* An SD 1.4 generation from the prompt "Walter White wall mural" extracted from a dataset of real-world prompts (DiffusionDB (poloclub)), alongside a corresponding source image containing copied elements. See Section 4.4 for details.

models may reproduce protected content or reveal sensitive information. As such models are embedded into everyday applications, users may encounter outputs that reuse data in unintended contexts, including images of individuals who did not anticipate or consent to such usage (Tramèr et al., 2024).

Recent work has therefore investigated how training data is memorized by generative models and how such memorization can be extracted, including unintentionally. Several studies (Somepalli et al., 2023b; Carlini et al., 2023; Webster, 2023) demonstrate that reconstruction attacks can recover training data in a blatant, near-verbatim manner. These attacks typically assume access to the training data, substantial computational resources, and prompts extracted from the data itself. As such, they simulate an explicit adversary attempting to recover memorized content, rather than the risk of unintentional reconstruction triggered by benign user prompts.

To better understand these risks, we propose an attack based on simple prompts referring to generic objects (e.g., t-shirt, beach towel), where access to exact training instances is replaced with informative priors about the training corpus. Applying this attack to previously studied models, we recover images containing elements whose sources can be traced, including images of real individuals. Prior attacks typically relied on highly specific prompts and aimed to reproduce a particular image. For example, prompting with a person's name such as "Ann Graham Lotz" to recover a known photograph (Carlini et al., 2023).

In contrast, our results show that potentially unintentional prompts can yield images of real people without naming them (see Figures 3 and 6).

Previous work largely focused on extracting training-set prompts and identifying duplications. In contrast, our attack builds on the inherent template-based text–image structure of a significant portion of the data, under the assumption that content scraped from certain websites is common in training corpora. For instance, e-commerce platforms produce many images that are nearly identical up to a fixed region where a print is overlaid, together with minor variations such as small zoom or crop differences or the addition of a shop logo, and are therefore likely present in large-scale web-scraped datasets such as LAION-5B (Schuhmann et al., 2022). While exact duplicates can sometimes be identified (Webster et al., 2023), such variants are often not flagged despite sharing substantial template-level visual structure, which we leverage to design an attack that does not require direct access to the training data.

Our contributions are threefold:

1. We propose a low-resource reconstruction attack for template-memorized images that operates in a black-box setting and requires no access to the training data, where prompts are constructed from real-world collocations instead of the training set.

2. To our knowledge, we are the first to demonstrate that memorized visual content can be reconstructed through seemingly benign prompts that may arise in normal model usage by naïve users. Hence, our attack uncovers a novel risk in text-to-image diffusion models not exposed by prior work which focused on anomalous prompts from the training data that are unlikely to be encountered in normal use.

3. Beyond reconstruction, we observe previously undocumented phenomena manifesting in template memorization, including interpolation, perturbations, and template leakage. These observations provide novel insights into how memorization manifests in text-to-image models.

## 2. Related Work

As the widespread use of generative AI models introduces increasing risks, several works actively study the extent of memorization in such models. In the context of LLMs, several attacks have been proposed demonstrating how data can be retrieved from the model (Carlini et al., 2021; Cooper et al., 2025; Nasr et al., 2025; Chen et al., 2024).

The issue of copying has also been observed in diffusion models with several implications, and ethical issues. For example, (He et al., 2025) identifies that diffusion models generate recognizable copyrighted characters from indirect or generic prompts.

More in line with our work, verbatim copies have also been generated in previous works (Somepalli et al., 2023a; Carlini et al., 2023; Webster, 2023; Webster et al., 2023; Wen et al., 2024). OpenAI (2022) identifies memorization by searching for prompts that yield generated images perceptually similar to training images, showing that memorized outputs often correspond to training samples with many near-duplicates. Building on this intuition, Carlini et al. (2023) explicitly assume that memorized images will exhibit duplication both in the training data and in generated outputs. Their method selects highly duplicated image–caption pairs from the training set. (Somepalli et al., 2023a) uses random sampling and study memorization by analyzing a random subsample of the training data. Webster (2023) further refine candidate selection by introducing a white-box filtering stage based on one-step denoising behavior. Starting from approximately 2M fully duplicated image–text pairs identified in LAION using their duplication metric (Webster et al., 2023). Subsequent works (Wen et al., 2024; Chen et al., 2025) do not propose new candidate extraction methods, but instead analyze the fixed list of 500 prompts released by Webster (2023) using additional white-box techniques to study memorization dynamics.

In contrast to all prior approaches, our method does not rely on access to training data, duplicated image–text pairs, or prompts extracted from the training set. Instead, we identify candidate prompts directly from websites, leveraging, as a weakness, the inherent template-based text–image structure of e-commerce websites. This enables a low-resource, black-box reconstruction attack and indeed our method generates orders of magnitude fewer images compared to previous work (Carlini et al., 2023; Webster, 2023).

Another feature of our method and attack is that it targets *template-memorized images*. These images are not full verbatim copies but may only contain elements from existing works. Such images become an important risk factor, as they are harder to detect by existing methods that often look for full verbatim copies (Carlini et al., 2021; Somepalli et al., 2023a;b; Wen et al., 2024). Such phenomena were identified in a few of previously constructed images Webster (2023); Somepalli et al. (2023a). In contrast to previous attacks, though, our attack targets specific, known, elements or objects such as t-shirts, curtains etc. and as such the phenomenon of memorized templates is much easier to identify and this allows us to further scale up the phenomenon and provide further analysis. As such, this phenomenon received lately specific interest with Di et al. (2026) proposing special mitigation techniques and,

concurrently with our work, (Chen et al., 2025) proposes a complementary method that targets local memorization.

# 3. Our Attack

## 3.1. Data Collection

Our attack, as in previous work, first builds on extracting potential prompts that will trigger copied works. A distinctive trait of our attack is that we avoid any access to the training set, and want to generate more natural prompts that will lead to reconstructed data. Because we avoid access to the training set, we cannot perform the partial-duplication search described. Instead, we compile a list of candidate expressions associated with e-commerce websites known to appear in LAION-5B. We obtain an initial set of consumer-facing print-on-demand websites using a broad query to ChatGPT, from which we extract product categories and object names corresponding to items offered for sale. Alternative website selection strategies yield similar sets of candidate expressions. The full list of websites used in this process is provided in Appendix B.

For each of these sites, we look at the snapshot of the site from March 2021, since the latest Common Crawl version to enter LAION 5B was April 2021. We then scraped a list of collocations from categories that are sold on these websites such as "unisex t-shirts", "athletic shoes" etc. Overall we reached a list of around 108 collocations. In e-commerce websites, images for each category are generated through different templates. Therefore, in order to generate diversity that will allow us to identify duplicates, we generated, for each collocation, a set of prompts by adding a description-visual pattern to each. The descriptions were: "Galaxy", "Floral", "Abstract Art", "I Heart ML", "blue", "red". These descriptors are intended to represent common visual descriptors appearing in e-commerce websites, including patterns, prints, and solid colors. This process leads to a set of prompts, pattern+collocation phrases, such as "Floral Unisex t-shirts". For each such prompt we generate $25-50$ different images (distinct by their seeds from 0 to 49).

**Collection from previous work** For direct comparison to prior work, we additionally extracted candidate collocations from full prompts previously identified in the literature. Analyzing these categories under our evaluation reveals qualitative patterns that closely mirror those observed in our own data collection, while also exposing recurring structures that, in some cases, correspond to copied content that was examined in earlier work but not identified as memorized. All details regarding the construction of these comparison collections are deferred to Appendix A.1.

## 3.2. Searching Near-Duplicates

Similarly to prior work (OpenAI, 2022; Carlini et al., 2023), we search for near-duplicate images among those generated by our pipeline. In our setting, images belong to known semantic categories, which allows us to estimate an editable region mask prior to similarity computation. This focuses the search on the fixed region and enables identification of cases that are near-identical up to the editable region.

We apply category-specific, pre-trained segmentation models to estimate the editable region mask, using Mask-Former (Meta AI, 2022) for household items and Seg-Former (mattmdjaga, 2024) for clothing. We then perform a clique search based on similarity of the fixed region, where edges connect image pairs with CLIP cosine similarity above $0.95$, a threshold selected empirically. Cliques of size at least two are flagged as suspected template-memorized cases, as the occurrence of even two near-identical images is unlikely under random generation, and are further filtered to identify near-duplicates.

Using CLIP similarity rather than $\ell_2$ distance allows inclusion of images with small perturbations that preserve overall structure but would evade strict pixel-wise comparisons (see Section 5.2). As a result, this approach recovers near-duplicate structures that are not detected by full-image similarity methods.

The method is limited by the accuracy of the editable region mask and may therefore yield partial results. We leave improvements to future work and complement the automated search with manual visual inspection to recover missed cases.

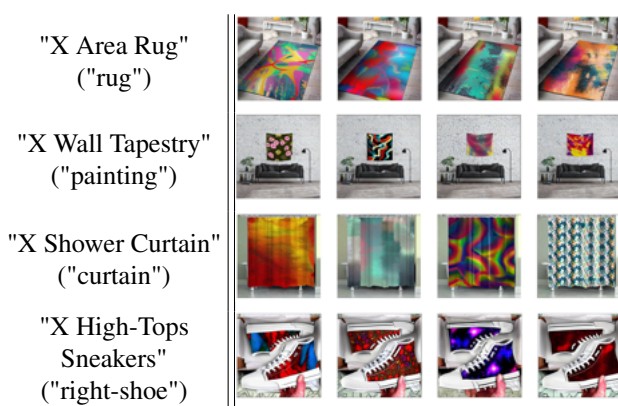

*Figure 2.* Examples of image cliques found through our segmentation and masking method. Segmentation category appears in brackets.

## 3.3. Tracing the Source Image

While for duplicated images we suspect that they are generally copied, for several of the images, we could validate and find a source. Sources were traced by Google Lens search, by targeting search to e-commerce websites and through visual inspection of images associated with the website categories.

As we could narrow down the search of the source to handful of websites, some images were recognized even without duplications, such as the generated image shown in first row of Figure 6. For this reason, we then continued to visually inspect the generated images comparing them to the original images in the origin websites, under their product categories, leading to the identification of additional memorized images.

| Source | Generated | Prompt |
|---|---|---|
|  |  | "Galaxy Area Rug" (source: GearFrost) |
|  |  | "Floral iPhone Case" (source: eBay) |
|  |  | "Abstract Art Shower Curtain" (source: Amazon) |

*Figure 3.* Examples of template-memorized images reconstructed through our attack on SD 1.4. For each pair, the source image (left), the generated image (center), and the corresponding text prompt (right) are shown.

## 4. Results

All of our experiments were conducted on a single RTX A6000 machine and, excluding those in Section 4.2, used Stable Diffusion version 1.4 from the official checkpoint accessible at HuggingFace (CompVis), which is the model where previous attacks were introduced (Carlini et al., 2023; Webster, 2023; Somepalli et al., 2023b). Results for more recent models are discussed and reported in Section 4.2.

The 108 collocations that we tested, of which 43 extracted at least one template, produced 67 image templates that we could identify. The dispersion of images is relatively uni-

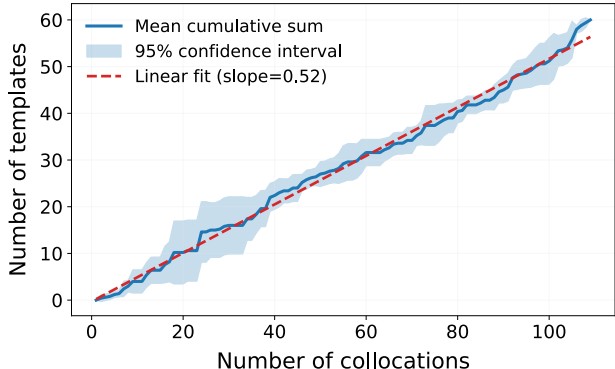

*Figure 4.* Number of collocations tested vs the number of identified image templates, mean of 5 permutations.

form, as summarized in Figure 4. In particular, our method of choosing collocations provides constant marginal returns. In total, we generated $11,400$ images—on par with the $9,000$ images produced by (Somepalli et al., 2023a), and several orders of magnitude fewer than the $175$ million generated in (Carlini et al., 2023), as summarized in Table 1. Unlike prior work, which treats each reconstruction as an isolated text–image pair, our analysis identifies underlying templates, capturing a broad space of variations enabled by the editable regions described above. As shown in Figure 5, this allows our method to more precisely identify prompts that reproduce copied content. We further validate our manual identification of copied images through a user study, summarized in Section 4.3.

| Paper | # Prompts | # Variations | # Copied |
|---|---|---|---|
| Carlini et al. (2023) | 350,000 | 500 | 109 |
| Somepalli et al. (2023a) | 9,000 | 1 | 170 |
| Webster (2023) | 30,000 | 500 | 153 |
| **Ours** | 108 | 100 | 67 |

*Table 1.* Comparison of generated prompts and identified copied images across prior works and ours. "Variations" denotes the number of seeds per prompt in prior works and the average seeds × descriptors per collocation in ours.

The full list of template groups that we identified, collocations and their associated image template, are provided at the appendix Section C.

### 4.1. Images With Identified Source

In Figure 2, we show examples of four cliques identified across different collocations. Since the cliques are formed using CLIP embeddings rather than raw $\ell_2$ distance, the grouped images may exhibit minor variations such as zoom, color, or pattern changes.

For many duplicated images, we were able to locate corre-

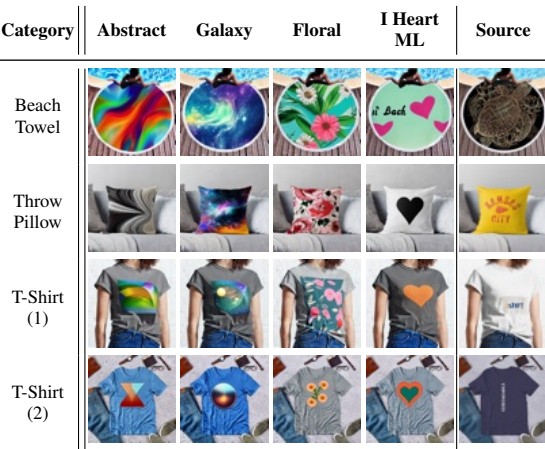

| Category | Abstract | Galaxy | Floral | I Heart ML | Source |
|---|---|---|---|---|---|
| Beach Towel | | | | | |
| Throw Pillow | | | | | |
| T-Shirt (1) | | | | | |
| T-Shirt (2) | | | | | |

*Figure 5.* Examples across product categories, showing different prompt themes (columns) and their corresponding real source image (last column).

sponding source images on the web, as shown in Figure 5. In addition, for some images not automatically identified as duplicates, we were still able to trace their origins by inspecting e-commerce websites from which our categories were derived, for example the image in the first row of Figure 6.

In Figure 13, we show two cases where the source images and their original captions could be identified in LAION, as they were borrowed from prior works with dataset access. The reconstruction prompts we used did not include these captions but rather generic descriptions, underscoring the risk of unintentional image conjuring.

**Real humans** Arguably, the most concerning type of image recall are those which potentially contain a real human model. Several of the duplicates we identified indeed contain real humans whose source images we could find, see Figure 6.

As discussed, previous works also managed to extract real-humans (Carlini et al., 2023). However this was generally done by prompts that actively request that person's image, and the focus was on copyright and verbatim copy. Here, the person is extracted by a prompt that, seemingly, does not contain their name, but instead the prompt contains a collocation associated with a product that the model had presented in the image. For example, mundane items such as "T-Shirt." This is both concerning as these models were trained to generate exactly such images, and because an uninformed user could generate such images under the impression that the image and the person in it are synthetic.

Even in cases where the face was deformed (as often happens in SD) or the picture is only of the torso, it might still contain identifying markers such as tattoos as well as other specific visual attributes such as haircut and pose. See Fig-

| Generated Image & Source | Prompt Used & Source |
|---|---|
| | "Abstract Art Unisex T-Shirt" (source: Redbubble) |
| | "Abstract Art Graphic T-Shirt Dress" (source: Redbubble) |
| | "Abstract Art Essential T-Shirt" (source: Zazzle) |

*Figure 6.* Examples of generated people that we could identify in a source (left: source, right: generated image). The first row contains a person identified by name (Wolf, 2018). Attack conducted on SD 1.4.

ure 10 (left columns) for further details.

### 4.2. Attacks on State-of-the-Art and Other Models

While we focused our efforts and designed our attack on Stable Diffusion 1.4 (SD 1.4), we could also extract template-memorized images from other popular generative models. Primarily, by using the same collocations identified in SD 1.4. In Figure 8 we exhibit further images extracted from *DeepFloyd IF-XL-I-v1.0* (DeepFloyd) and Midjourney V4 (Midjourney). Both models were released around the time of SD 1.4, and in Figure 7 we demonstrate further extractions from state-of-the-art (SOTA) models.

- **DeepFloyd IF-XL-I-v1.0**: although the generated images exhibited lower contrast, which made clique search impractical, we successfully identified template-memorized images through visual inspection. In addition, we discovered new templates, absent from SD 1.4, using Google Lens searches on manually flagged generations.

- **Midjourney V4:** MidJourney (Midjourney) posed additional challenge for our attack, due to the lack of an open-source implementation or public API. However, by restricting our experiments to a limited set of prompts and seeds, selected based on template-memorized prompts identified in SD 1.4, we could still apply our attack.

For each prompt and seed, we collected the four de-

**Stable Diffusion v3.5**   **Flux-Schnell v1.0**   **MidJourney v6.1**
Source      Generated        Source      Generated        Source      Generated

*Figure 7.* Template-memorized images extracted from state-of-the-art models. For each model, the source image (left) and corresponding memorized generation (right) are shown.

fault outputs returned by Midjourney, specifying version 4 with the argument `-v 4`. Despite limited prompt and seed control, we successfully extracted template-like generations from Midjourney. Our results included both direct template replications from SD 1.4 and interpolated reconstructions (see Section 5.1)).

While the training data for Midjourney remains undisclosed, given the scale of the model and industry norms, we assume that large portions of the training corpus were sourced from publicly available internet data.

**DeepFloyd**
Source   Generated   Source   Generated   Source   Generated

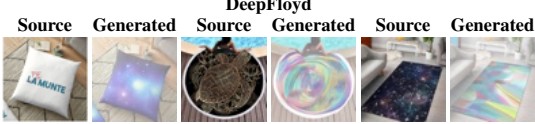

Example results from our attack on **DeepFloyd**. For each pair, the source image (left) and the generated image (right) are shown.

**Midjourney v4**
Source   Generated   Source   Generated   Source   Generated

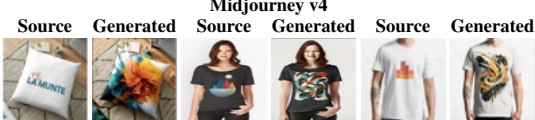

*Figure 8.* Reconstructed images from models of the same generation as SD 1.4. For each pair, the source (left) and generated (right) images are shown.

- **SOTA Models:** In addition to attacking models from the same generation of SD, we further evaluated the vulnerability of state-of-the-art (SOTA) text-to-image models using collocations identified as template-memorized in SD 1.4. We attacked the following models: *Stable Diffusion 3.5 Medium* (Stability AI), *Flux Schnell v1.0* (Black Forest Labs), and *Midjourney v6.1* (Midjourney).

For Stable Diffusion and Flux Schnell we used the Hugging Face checkpoint in the same manner we've conducted our main experiments on SD v1.4, while for Midjourney we used the Discord interface as was detailed previously. We successfully demonstrated

some memorization, as seen in Figure 7, though to a lesser extent than in older models and versions.

**SD 3.5** introduced partial mitigation techniques targeting image-text coupling vulnerabilities; however, these measures were not specifically designed to address the template-style coupling exploited by our attack, which relies on concise prompts derived from external sources rather than training data. Our results indicate that SD 3.5 exhibits increased resilience but remains partially vulnerable to template memorization. Representative examples are provided in Figure 7

**MidJourney V6.1**, as discussed, poses additional challenges. However, we applied the same methodology used for V4. While the generated images were typically perturbed (see Section 5.2), targeting previously vulnerable benign prompts still yielded images with notable resemblance to the source. For instance, as shown in Figure 7, the prompt "Abstract Art Essential T-Shirt" (which previously produced near-verbatim copies) continues to generate closely resembling images.

### 4.3. User Study

To validate our identification of copied content, we conducted a user study evaluating whether the copying we observed is visually apparent to human observers. As shown in Figure 9, user judgments strongly align with our annotations across direct template memorization, interpolations, and perturbations. Full details of the study design and analysis are provided in Appendix A.4.

### 4.4. Our Prompts In The Wild

We previously characterized our attack as reflecting the behavior of a naïve user who interacts with the model exactly as intended. To further support this claim, we analyze two large-scale datasets of real user prompts (vivym; poloclub), collected directly from everyday model usage.

First, in each dataset, we search for prompts containing our identified collocations to validate that indeed such colloca-

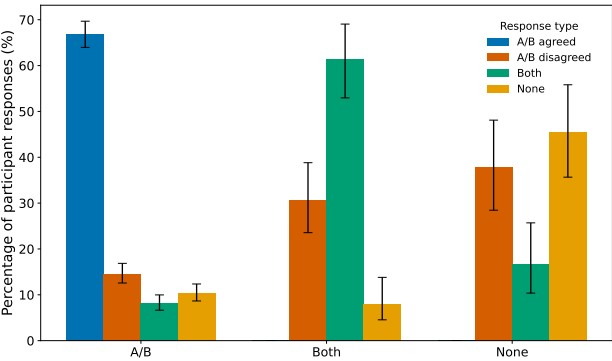

*Figure 9.* User study results aggregated by our ground-truth labels with cross-participant CI.

tions appear in every-day use. In the Midjourney dataset (vivym), we managed to find 12,706 total prompts (3,883 unique) containing these collocations out of 7.13M samples. In the Stable Diffusion dataset (poloclub), we could find 818 total prompts (660 unique) out of 1.8M samples.

The original images generated from these prompts were unavailable due to link expiration (Midjourney) and dataset scale (DiffusionDB) so we could not estimate how many of these prompts actually yielded TMIs in real-world use cases. Nevertheless, as a preliminary assessment, we conducted a small, low-resource experiment focusing on a limited set of selected candidate prompts. From the 3,883 unique Midjourney prompts and 660 unique DiffusionDB prompts that contain our collocations, we select only the shortest prompt, yielding curated subsets of 35 candidate prompts. We then generated images using our previously described pipeline and manually inspect them for the templates we have identified (see Section C). Despite the modest scale of this experiment, the results were notable: 8 out of 20 Midjourney prompts and 7 out of 15 DiffusionDB prompts produced TMIs. An example prompt and its resulting TMI appear in Figure 1. Although preliminary, these findings suggest that TMIs may be more common in the wild than previously assumed. A more extensive study is needed to determine their true prevalence and evaluate the seriousness of this phenomenon.

### 4.5. Evaluation of BE-PRSS Detection and Mitigation

We evaluated the detection and mitigation methods proposed by Chen et al. (Chen et al., 2025) on our template-memorized examples. We constructed prompts by adding the descriptor "Abstract Art" to each tested collocation, and ran each prompt with 3 different seeds using the BE-PRSS detection script. We manually annotated each generated image as TMI or non-memorized by comparing it to the corresponding known template. Some outputs annotated as TMI were interpolation cases, as described in Section 5.1.

Using the default BE-PRSS threshold of 1.0, the detector reached an accuracy of 0.595, precision of 0.520, and recall of 0.722. Thus, while the method detected some TMI cases, it was not reliable for template-level memorization in our setting.

We also tested the mitigation method on the 43 collocations previously associated with memorization. For each collocation, we generated 25 prompt variations using ChatGPT, as required by the mitigation script, and counted whether at least one of three seeds produced a TMI. Before mitigation, 28 collocations produced a TMI; after mitigation, 22 did. This suggests only a modest reduction. We also observed utility degradation, where some mitigated prompts no longer generated the requested item. We believe this is due to the method's reliance on prompt variations, which assumes memorization is tied to a specific prompt, whereas our results suggest a many-to-many structure driven by collocations or keywords rather than the full prompt.

## 5. Observed Phenomena in Template Memorized Images

Analyzing a large set of reconstructed template-memorized images, we observe three recurring phenomena: **Interpolation**, **Perturbations**, and **Leakage**, which, to the best of our knowledge, have not been previously reported.

We further find that template memorization follows a many-to-many structure, which we refer to as **template groups**, where multiple collocations correspond to multiple image templates rather than a one-to-one text–image association. Concurrent work (Di et al., 2026) has recently observed a one-to-many structure, in which a single prompt gives rise to multiple related generated images; however, this does not capture the full many-to-many organization we identify. While related structure was hinted at by (Webster, 2023), it was neither explicitly defined nor systematically analyzed. Collocations within a template group are either semantically related (e.g., duvet cover, comforter, throw blanket) or co-occur in product descriptions as alternative variants of the same item (e.g., picnic blanket, beach towel). The identified template groups are listed in Appendix C.

### 5.1. Interpolation

The first phenomenon we observe is interpolated reconstruction, in which generated images blend elements from multiple training examples rather than copying any single image outright. Such partial copying often evades standard similarity metrics: neither the background nor the foreground is directly duplicated, yet individual elements can be readily matched to their respective sources by a human observer, as further validated in the user study presented in

Section 4.3. For example, in Figure 10, one image reproduces a tattoo from a real-world photograph while borrowing hair from another source. The combination of elements from multiple images raises challenging questions regarding what constitutes a memorized output.

This behavior suggests a deeper mechanism, whereby diffusion models internalize, reorganize, and recombine fine-grained visual traits drawn from multiple training images. In our experiments, we identify interpolated reconstructions that clearly fuse elements from two and three distinct memorized sources.

We further speculate that some generations may blend subtler fragments from a broader set of images, producing composites whose lineage spans many sources. If such multi-source blending indeed occurs, attributing influence to specific training images becomes increasingly difficult, sometimes effectively impossible. This speculation raises philosophical questions about whether these models can be said to generate truly novel samples, or whether every output is in fact a rearrangement of training data.

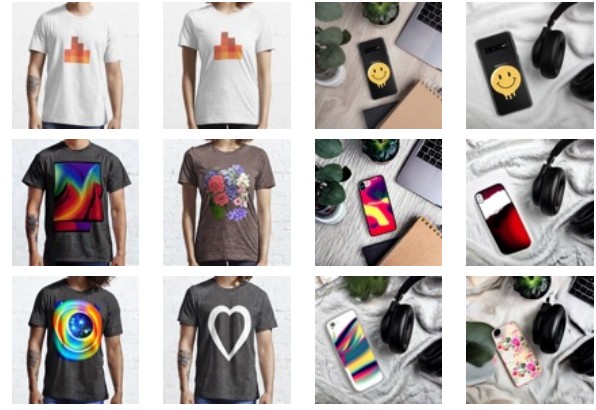

*Figure 10.* Examples of interpolations we observed. Top row: source images. Middle row: generated templates (TMI). Bottom row: interpolations. Left: "Essential T-Shirt", right: "iPhone Case & Cover".

### 5.2. Perturbations

We also observe clusters of images that are nearly identical in their underlying template, yet differ through localized substitutions of semantically similar objects placed in the same position—for example, replacing one lamp with another or swapping between similar chair designs. An example of this behavior appears in Figure 11. These perturbations preserve high CLIP similarity on the fixed region, while introducing small pixel-level differences that modestly reduce $\ell_2$ similarity, though the images remain far more similar than random pairs.

The origin of this phenomenon is difficult to isolate, but it aligns with two plausible explanations. First, small pertur-

bations to the latent noise can induce localized visual variations in the final image, consistent with behavior demonstrated in (Stenbit et al., 2022). Second, it may reflect the one-step denoising behavior described by (Webster, 2023), which we observe specifically in memorized images: once the global layout is recovered in the initial step, subsequent denoising allows only limited variation, resulting in small but systematic perturbations across generations.

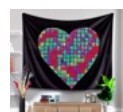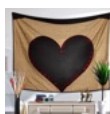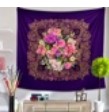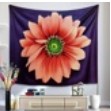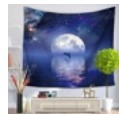

*Figure 11.* Four images with perturbations (see plant on the right) generated from the prompt "X Wall Tapestry", where $X \in$ {"I Heart ML", "Floral"}. Rightmost image is the source found via Google Lens.

### 5.3. Leakage

A third phenomenon we identify is *template leakage*, in which an image template associated with one template group appears in generations prompted from another group or from a prompt not associated with any group. In such cases, the reused template typically overlaps with the edited region in a visually coherent manner. An illustrative example is shown on the right of Figure 12, where a template strongly associated with the "T-Shirt" group appears in a generation prompted for a "Tank Top." While the edited region correctly depicts a tank top, the surrounding background matches the T-shirt template observed in both the source image and other reconstructions.

At the same time, inspection of e-commerce websites shows that some templates are legitimately reused across product categories, meaning that not all suspicious cases necessarily indicate true leakage. For example, the left side of Figure 12 shows a background shared by both "Canvas Wall Art Print" and "Wall Tapestry," reflecting genuine reuse of the same template in the training data.

While template leakage cannot be proven conclusively due to such reuse, we support its existence by intentionally inducing leakage in controlled synthetic settings, as detailed in Appendix A.3.

## 6. Discussion

This work highlights that template-level memorization can manifest in ways that evade existing detection methods, particularly through interpolation, perturbations, and cross-category leakage, where copied content is not captured by standard similarity metrics. This limitation stems from the fact that commonly used metrics (e.g., $\ell_2$, SSCD) operate on full-image, pairwise comparisons, where the dominant signal often comes from the editable region, while

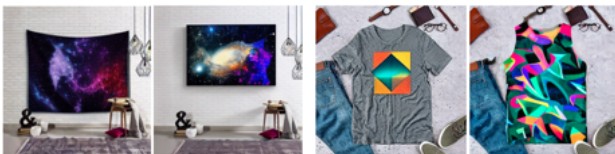

*Figure 12.* Examples of template sharing and suspected cross-category leakage. **Left:** Background shared between template groups, not suspected for leakage, as sources were found for both prompts ("Galaxy Wall Tapestry" and "Galaxy Canvas Wall Art Print"). **Right:** Suspected leakage from "T-Shirt" to "Tank Top": a source was found for "Abstract Art Essential T-Shirt", while no source was identified for "Abstract Art Tank Top".

the copied template occupies only a localized subset; moreover, in interpolation settings, copied elements may originate from multiple sources, further reducing pairwise similarity to any single reference. While avoiding access to the training data better reflects realistic model usage and downstream risk, it also limits the precision with which memorized samples can be identified and attributed. As a result, our analysis relies on manual inspection and domain knowledge, which currently constrains scalability. Developing automated and scalable methods for detecting non-verbatim, template-level memorization remains an important direction for future work, both for understanding memorization mechanisms and for mitigating unintended reuse in deployed generative systems.

## Impact Statement

This work studies memorization phenomena in text-to-image diffusion models and demonstrates a risk that arises during everyday model usage by naïve users, rather than in adversarial or purely theoretical settings. We show that benign, naturally occurring prompts can lead to the reconstruction of memorized visual content through standard model interfaces, highlighting a gap between existing threat models and real-world usage.

The goal of this research is to promote a clearer understanding of how memorization-related risks manifest in practice, and to support the development of improved evaluation, mitigation, and data curation strategies. By characterizing these behaviors empirically, this work aims to inform model developers, practitioners, and policymakers about potential failure modes that may otherwise go unnoticed.

The methods presented do not provide new access to training data beyond what is already exposed through normal model interaction, and we do not release reconstructed personal data or tools intended for large-scale exploitation. Instead, this work emphasizes responsible disclosure and the importance of addressing memorization risks as text-to-image models continue to be widely deployed.

Overall, we believe that identifying and contextualizing risks that already occur in real-world usage is a necessary step toward safer and more trustworthy deployment of generative models.

**Acknowledgment** This work has received funding from the European Research Council (ERC) under the European Union's Horizon 2020 research and innovation program (grant agreement FoG-101116258). Views and opinions expressed are however those of the author(s) only and do not necessarily reflect those of the European Union or the European Research Council. Neither the European Union nor the granting authority can be held responsible for them. This work received additional support from the Tel Aviv University Center for AI and Data Science (TAD) and a grant from the Israeli Council of Higher Education.

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

| Generated Image | Source Image | Prompt and Source Caption |
|---|---|---|
|  |  | **prompt**: Galaxy Print Universal Fit Car Seat Covers.
**Caption**: Wild Hearts Can'T Be Broken Car Seat Covers For Horse Lovers 170804 - YourCarButBetter |
|  |  | **Prompt**: Abstract Art Round Metal Wall Art.
**Caption**: Designart 'Wide Pathway in Yellow Fall Forest' Landscape Photo Round Metal Wall Art . |

*Figure 13.* Comparison of Generated and Source Images with Corresponding Prompts and Captions when taking the categories from previous works, examples from (Hintersdorf et al., 2024).

## A. Additional Results

### A.1. Comparison with previous attacks

As discussed, to allow comparison with previous attacks we additionally extracted a list of collocations derived from memorization cases reported in prior work. Rather than harvesting generic e-commerce websites, these collocations were isolated from full prompts previously identified as memorized, focusing on product-related descriptions. This results in a set of collocations that is comparable in nature to those extracted from the generic websites used in our data collection, while being grounded in sources explicitly analyzed in earlier attacks.

Concretely, given full prompts reported in prior work, we manually extracted short product-describing phrases (typically one to three words), such as extracting the collocation *Car Seat Covers* from the prompt "Tribal Aztec Indians pattern Universal Fit Car Seat Covers" appearing in the list of prompts published by (Webster, 2023) as passing their black box attack, i.e. not template-memorized. This enables a direct comparison between our approach, which relies on natural prompt construction without access to the training set, and prior approaches that extract or analyze captions originating from the training data. Figure 13 presents example generations and the corresponding prompts used in each setting.

Demonstration of the one-step denoising behavior presented at (Webster, 2023) can be seen at Figure 14.

### A.2. Traces of Memorization in Stable Diffusion 3.5

Beyond the clear instances of template extraction—sometimes preserved only up to perturbations, as shown in Figure 7—we also found that Stable Diffusion 3.5 exhibits subtler forms of memorization. Even when our original attack no longer yielded a recognizable reconstruction, meaningful traces of the underlying training content persisted, suggesting that future, more targeted attacks may still uncover sensitive material.

These findings highlight the significance of the phenomena described in this work—interpolations, perturbations, and template leakage. Without understanding these behaviors, one might incorrectly assume that failed reconstructions imply the absence of memorization. Instead, these phenomena reveal that memorized content can manifest in isolated visual elements, partially altered fragments, or blended structures rather than as a direct or verbatim template. In other words, the form in which memorization appears can change—even when its presence does not.

A concrete example appears in Figure 15. Although the image template had no longer been fully reconstructed in Stable Diffusion 3.5, its generation still exhibited recognizable elements, such as a person in a specific pose. Using the template groups we had previously identified for Stable Diffusion 1.4, we were able to match this distorted output back to the same underlying template and corresponding source image. This shows that, even when verbatim extraction is no longer possible, remnants of training images, particularly from large-scale e-commerce data,can survive as structured traces tied to known templates, and thus may continue to pose a risk in future model versions.

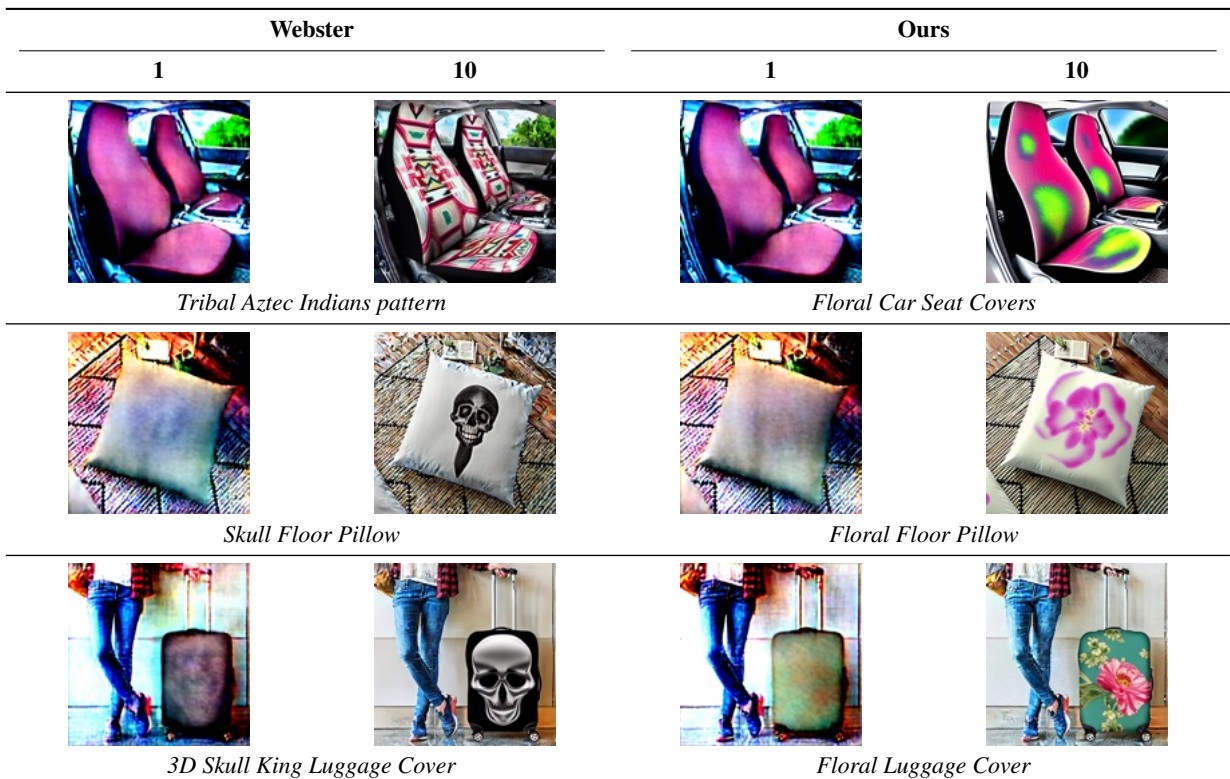

*Figure 14.* Comparison between images generated after 1 and 10 steps for Webster's attack and ours. From top to bottom: (a) identified by Webster's white-box only, (b) identified as non-verbatim, (c) identified as template verbatim.

### A.3. Synthetic Experiments

To better understand how the coupling between *text templates* and *image templates* contributes to partial memorization, we designed a series of synthetic experiments that deliberately recreate this form of coupling. Our goal was twofold: (1) to intentionally induce memorization under controlled conditions, and (2) to investigate another phenomenon we call *template leakage*. In real-world data, we frequently encountered images where a template from one category appeared under a different category. However, because e-commerce platforms often reuse templates across product types, it was impossible to determine whether these cases reflected true leakage or simply the natural reuse of templates in the training data. Synthetic experiments offered a way to isolate these effects.

**Stage 1: Controlled overlays with crude template replacement.** For the first stage, we collected three photographs of a coffee mug next to an iPhone SE, placing the same mug in three different locations in our lab. We then created manual masks for each image using Photopea and replaced the masked regions with simple patterns using OpenCV. At this stage, the overlays were intentionally crude—visually unrealistic and easy for the model to memorize.

Under these conditions, we observed *verbatim template extraction*, as expected, along with clear instances of *interpolation*, *perturbations*, and *leakage*. The model reproduced not only the underlying structure of the templates but also combined and varied them in ways that matched the phenomena we identified in real-world data.

**Stage 2: Realistic mockups mimicking Print-on-Demand rendering.** In the second stage, we constructed a more realistic setup based on Print-on-Demand (PoD) workflows. We selected three mockups from Freepik and replaced their smart-object contents in Photoshop using an automation script, with light manual adjustments to maintain uniformity across images. For one mockup, we additionally inserted two decorative elements—a pair of leaves and a lemon slice—also from Freepik, to mimic the kind of compositional variability seen in real product designs.

With these more naturalistic overlays, we no longer observed verbatim template extraction. However, *interpolation*, *perturbations*, and *object memorization* remained evident, confirming that these behaviors persist even when the templates are

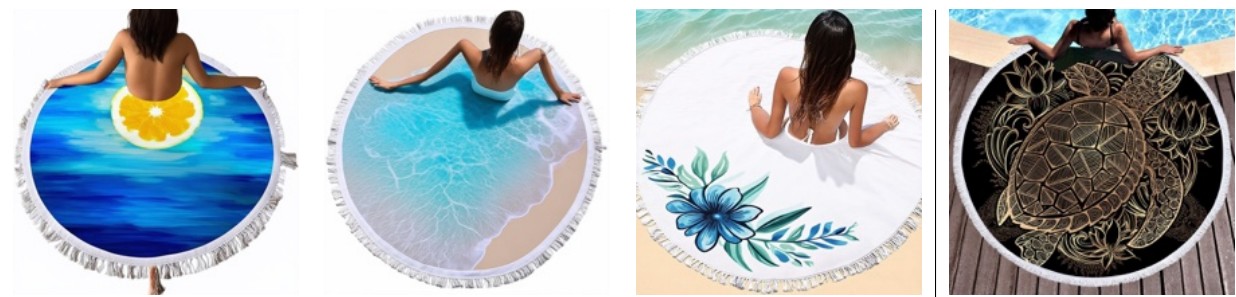

*Figure 15.* Traces of our attack in SD 3.5 (Medium). The first three images were generated using the prompt *"BOYOUTH Round Beach Towel,X Beach Mat with Tassels Ultra Soft Super Water Absorbent Multi-Purpose Towel,59 inch-Diameter"* Taken from the source image's product description in Amazon, where X was: "Abstract Art", "Floral", "Galaxy". The image to the right of the vertical line is the source image, which was reconstructed in previous models (see Figure 5)

more realistically embedded and less trivially memorized.

**Testing for template leakage under semantic shifts.** To probe template leakage directly, we generated images not only for the collocation "Coffee Mug" but also for prompts that were *semantically similar* ("Tea Cup") and *semantically dissimilar* ("T-Shirt"). These generations allowed us to separate genuine leakage from simple template reuse.

The results provided clear evidence of genuine leakage. Under prompts such as "skg Tea Cup," the backgrounds consistently showed interpolations—or, at minimum, strong color and texture echoes—of the coffee-mug templates used during fine-tuning. Even more strikingly, the generated tea cups appeared in *top view*, a viewpoint that never appeared in the training data and was absent from all coffee-mug generations, which only reproduced the three canonical views present in the training set. The cups also manifested in shapes not seen during training.

Together, these findings demonstrate not only that template leakage occurs, but also that synthetic experiments enable us to disentangle leakage from benign template reuse - revealing how template structures can transfer across prompts and categories under controlled conditions.

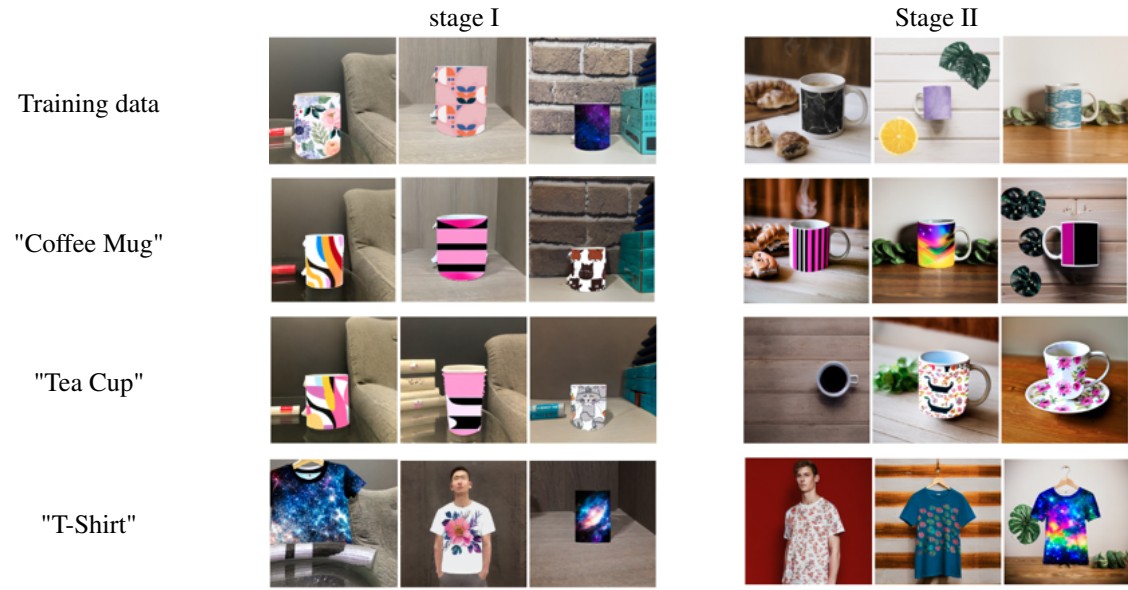

*Figure 16.* Intentionally causing template memorization by fine tuning SD on coupled image-text pairs. The generated results demonstrate the phenomena of interpolation, perturbations, and leakage.

## A.4. User Study

To validate our manual annotations of copied images, we conducted a user study to assess whether the identified copying is visually apparent to human observers. Prior work primarily focused on verbatim copies, where automated metrics such as SSCD or edge-based consistency can be incorporated into the detection pipeline. In contrast, our work targets template-level memorization, where reproduced content may appear in the background or differ in minor structural details. In this setting, the notion of copying becomes ill-posed, and we observed that edge consistency frequently failed to flag images that were nonetheless clearly copied to a human observer, motivating a user-based validation.

Participants were shown generated images alongside potential source images and were asked to identify copied content. The evaluated examples included both direct instances of template memorization (TMI) as well as cases of interpolation and perturbation (see Sections 5.1 and 5.2). Overall, the study comprised 56 source images with corresponding generated counterparts. We collected responses from 70 participants, each answering 18-19 questions.

For each generated image, we additionally selected a control image produced using the same prompt but a different randomly chosen seed. We also included questions in which **both** generated images or **neither** contained copied elements from the source. In each question, participants were shown a **triplet** consisting of one source image and two generated images and were asked:

> *"For each image pair, select which generated image appears to have elements copied from the source."*

The available choices were A, B, Both, and None. The questionnaire was administered via Google Forms with non-expert volunteers recruited through personal networks, and we recorded age and gender information to assess sample diversity.

We calculated the pairwise similarity between each image and the source image $s(source, TMI), s(source, control)$ used in previous works and compared them to the percentage of users who identified the image as having elements copied from the source. The results are shown in Figure 17. It is important to notice that each pair the image and the source were generated from the same prompt, which biases the similarities upward.

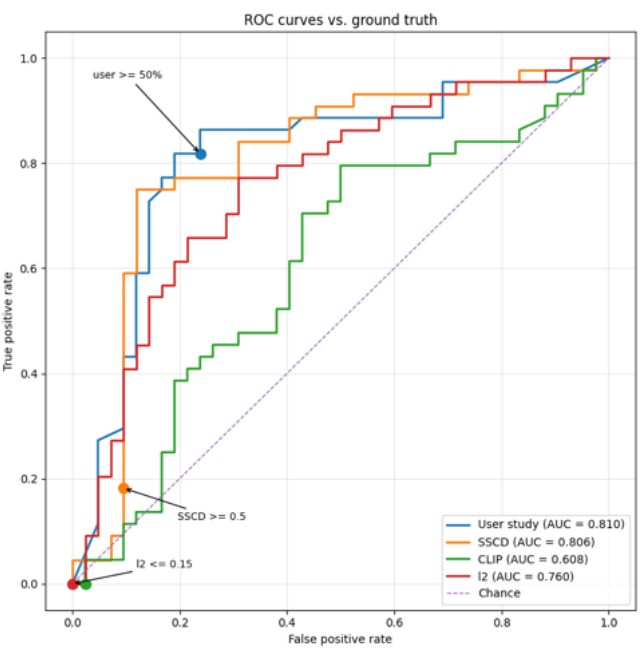

*Figure 17.* Comparison of similarity metrics with the user study, the thresholds marked on the graph are user study majority vote, SSCD from (Somepalli et al., 2023a), and $\ell_2$ from (Carlini et al., 2023)

## B. List of e-commerce websites

We list the consumer-facing print-on-demand and e-commerce websites used to construct candidate collocations in our experiments. This list was fixed prior to running the experiments and used consistently throughout all evaluations. We found that alternative website selection strategies produced largely overlapping sets.

- Redbubble (redbubble.com)

- TeePublic (teepublic.com)

- Zazzle (zazzle.com)

- Society6 (society6.com)

- Spreadshirt (spreadshirt.com)

- DesignByHumans (designbyhumans.com)

- CafePress (cafepress.com)

- Fine Art America (fineartamerica.com)

- Threadless (threadless.com)

- Printful (printful.com)

## C. Identified Template Groups

Here we list the template groups that we identified through our method described in Section 3.

For convenience, the collocations are grouped by product category and the categories are ordered alphabetically.

Under each collocation or set of collocations we show the different image templates associated with that collocation.

Some image templates are associated with multiple collocations as detailed in Section 5 thus appearing more than once. Also some image templates are similar but not exactly the same, we show them separately since they appeared consistently in both generated images and the source. For example the leftmost two image templates associated with "Wall Tapestry" vary by the ampersand shaped decoration and angle.

## Area Rug

Collocations: Area Rug, Rug, Floor Mat

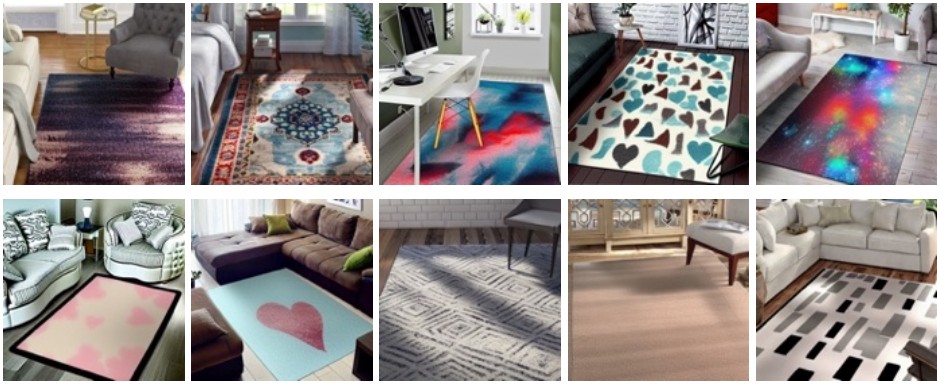

## Bag

Collocation: Backpack

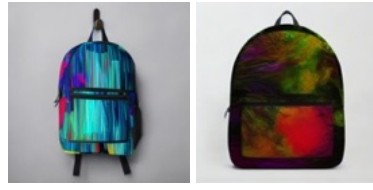

## Beach Towel

Collocation: Beach Towel

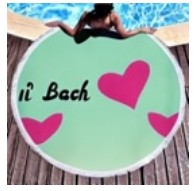

## Blanket

Collocation: Comforter

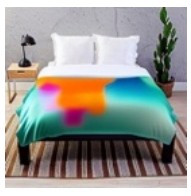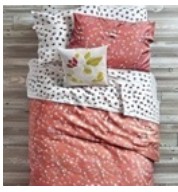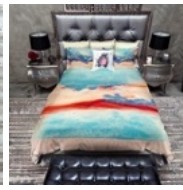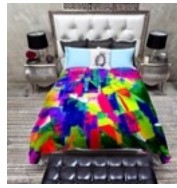

Collocation: Duvet Cover

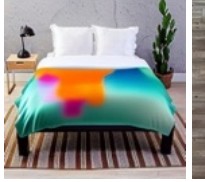 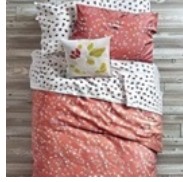

Collocation: Throw Blanket

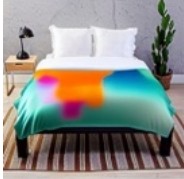 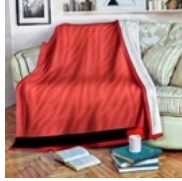 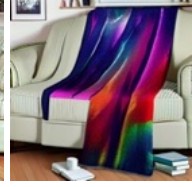

Collocation: Picnic Blanket

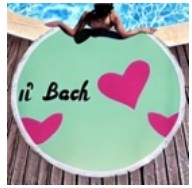 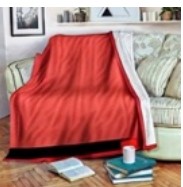

## Coasters

Collocations: Coasters (Set of 4)

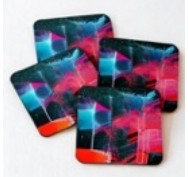

## Dress

Collocation: A-line Dress

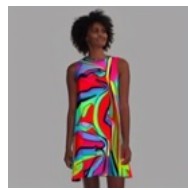 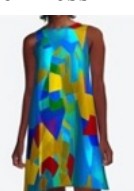

Collocation: Graphic T-Shirt Dress

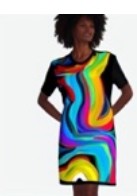 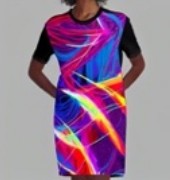

## Shoes

Collocation: High-Top Sneakers

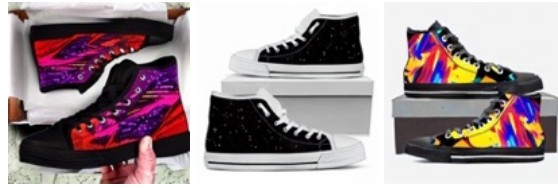

Collocation: Low-Top Sneakers

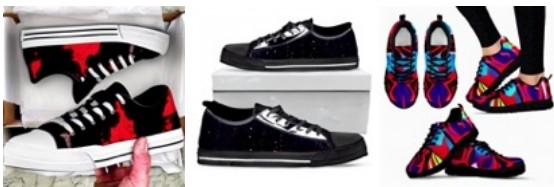

## Shower Curtain

Collocation: Shower Curtain

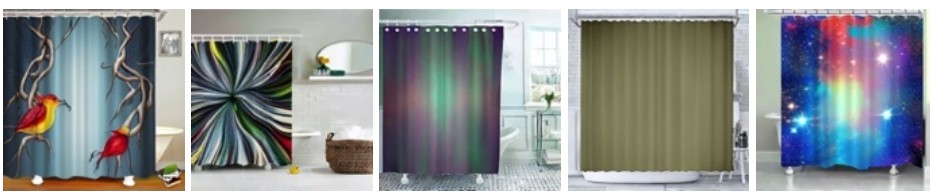

## T-Shirt

Collocations: T-Shirt
Including the phrases: Active T-Shirt, Baby T-Shirt, Baseball ¾ Sleeve T-Shirt, Classic T-Shirt, Essential T-Shirt, Fitted T-Shirt, Premium Scoop T-Shirt, Relaxed Fit T-Shirt, Tri-Blend T-Shirt,

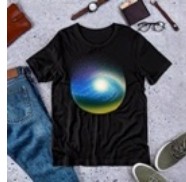

Collocation: Essential T-Shirt

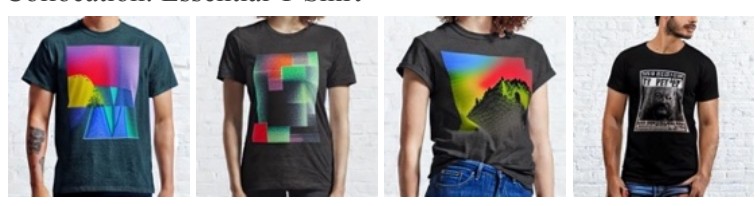

Collocations: Classic T-Shirt, Unisex T-shirt

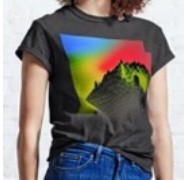

Collocation: Unisex T-Shirt

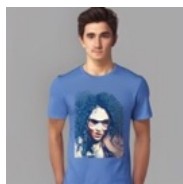

## Laptop and Tablet

Collocation: Laptop Sleeve

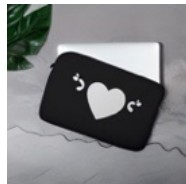

Collocation: iPad Case & Skin

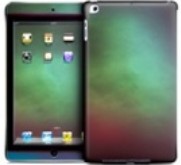

## Luggage

Collocation: Luggage

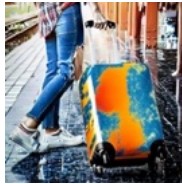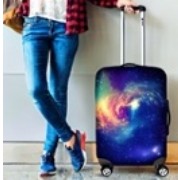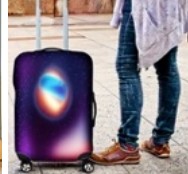

## Phone Case

Collocations:  Case & Skin for Samsung Galaxy, iPhone Wallet, iPhone Case & Cover
(This template appears in all of them), iPhone Case, iPhone Cover, Galaxy Case, Galaxy Cover

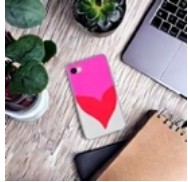

Collocation: Case & Skin for Samsung Galaxy

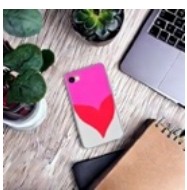 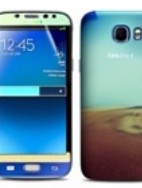 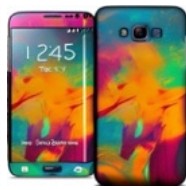

Collocation: iPhone Case & Cover

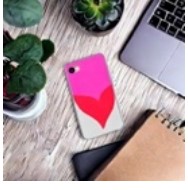 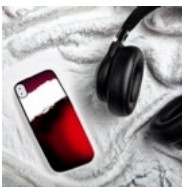 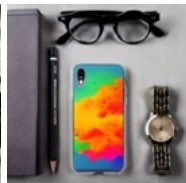 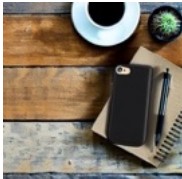

## Pillow

Collocation: Floor Pillow

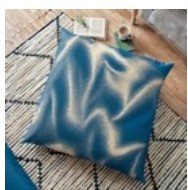

Collocations: Throw Pillow

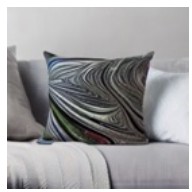 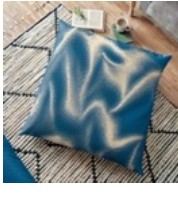 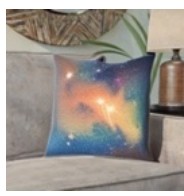 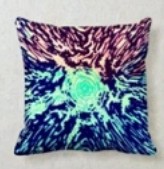 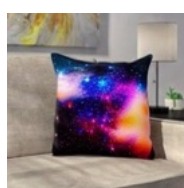

Collocations: Pillow Cover, Pillow Sham, Rectangular Pillow

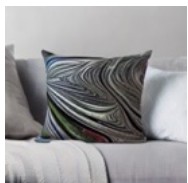 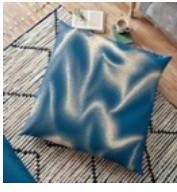

# Tapestry

## Collocations: Tapestry, Wall Hangings

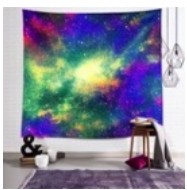

## Collocation: Wall Tapestry

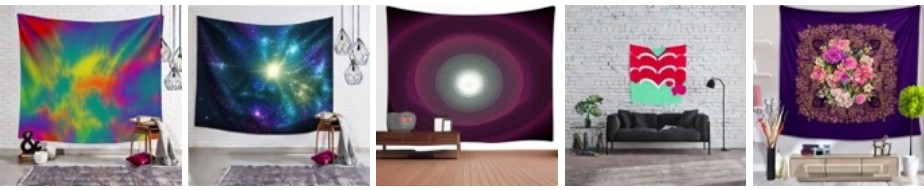

## Collocation: Wall Mural

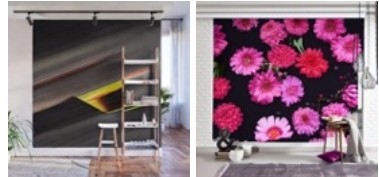

# Art Prints

## Collocations: Art Board Print, Mounted Print

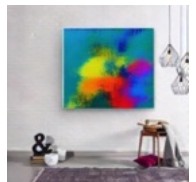

## Collocation: Framed Art Print

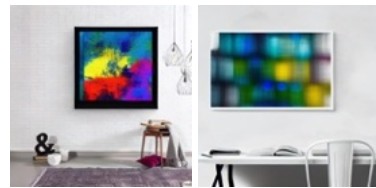

## Collocation: Canvas Print

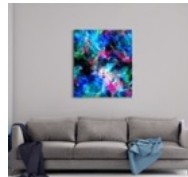