# OpenReview forum: "Reconstructing Template-Memorized Images from Natural Prompts"
_ICML.cc/2026/Conference — ICML 2026 regular_

### Official Review · Reviewer_zj6c · 2026-03-09

**Soundness:** 2
**Presentation:** 3
**Significance:** 3
**Originality:** 3
**Overall Recommendation:** 4
**Confidence:** 3

**Summary:**

This paper aims at finding template-memorized images (TMI), where recurring layouts and visual structures are memorized during training. They introduce a low-resource reconstruction attack that operates through seemingly benign prompts and requires little to no access to the training data. They demonstrate that memorized visual content can be reconstructed by normal model usage by naïve users. Furthermore, they observe novel phenomena occurring in TMIs, including interpolation, perturbations and template leakage.

**Compliance With Llm Reviewing Policy:**

Affirmed.

**Final Justification:**

I will keep my positive rating.

**Key Questions For Authors:**

1.	What does the proposed attack perform under some mitigation methods like Di et al. (2025a)?

2.	Do such template-memorized images exist and be able to be discovered by your method for modern commercial models like GPT, Grok, Gemini, etc.?

3.	How many reconstructed images were verified against training datasets (e.g., LAION) rather than external web search?

4.	What is the failure rate of the segmentation-based editable region masking, and how does it affect duplicate detection accuracy?

**Limitations:**

Yes.

**Strengths And Weaknesses:**

reconstructed by normal model usage by naïve users. Furthermore, they observe novel phenomena occurring in TMIs, including interpolation, perturbations and template leakage.

Strengths:

1.	Studying on Template-Memorized Images for modern text-to-image models is interesting and cool.

2.	The proposed attack is easy to be deployed in real-world scenarios without the need of leakage on training datasets, posing a practical threat model.

3.	Experiments studied on various models demonstrate the findings.

4.	Three interesting phenomena for TMIs, interpolation, perturbations and leakage are discussed.

Weaknesses:

1.	It lacks of explanations on why choose descriptions like “Galaxy”, “Floral”, “Abstract Art”, “I Heart ML”, “blue”, “red”. It is hard for me to find any relationship from these descriptions.

2.	Identification of Template-Memorized Images seems a little haphazard. In your proposed attack, TMIs are identified through segmentation and masking methods to search near-duplicates. However, no formal description exists for your clique search algorithm. Furthermore, when applying it in the wild, you cast away them and just enjoy human judgement. Moreover, looking at TMIs shown in Appendix C, I am not sure whether these images are actually similar. For instance, 10 TMIs for Area Rug looks quite different. Furthermore, source tracing relies on Google Lens searches and manual inspection. Lack of strict or comprehensive evaluation somewhat undermines the soundness of your findings.

3.	The similarity threshold for CLIP cosine similarity (0.95) is selected empirically without justification or ablation.

4.	Experimental scale is limited. Only 112 collocations are tested, In-The-Wild evaluation uses only 35 candidate prompts.

---

> ### Author Rebuttal · Authors · 2026-03-30
>
> ## Weaknesses:
>
>  **It lacks of explanations on why choose descriptions like “Galaxy”, “Floral”, “Abstract Art”, “I Heart ML”, “blue”, “red”**
>
> We chose descriptions that represent common visual descriptors appearing in e-commerce websites, including patterns (galaxy, floral), prints (abstract art, I heart ML), and solid colors (red, blue). We did not specifically curate these descriptors, and we expect that many common e-commerce patterns would induce the same phenomenon. We will clarify this in the paper.
>
> **In your proposed attack, TMIs are identified through segmentation and masking methods to search near-duplicates. However, no formal description exists for your clique search algorithm.**
>
>  For clique search we used an off-the-shelf algorithm from the Python library NetworkX. The implementation is available in the supplementary materials, and we will add these details to the manuscript for clarity.
>
> **when applying it in the wild, you cast away them and just enjoy human judgement.**
>
>  The goal of the “in the wild” section was not to recreate the full attack pipeline, but to show that the risk we identified also appears in real usage. We therefore used the image categories and potential targets already identified in the main attack.
>
> **looking at TMIs shown in Appendix C, I am not sure whether these images are actually similar. For instance, 10 TMIs for Area Rug looks quite different.**
>
> Sorry for the misunderstanding. Appendix C shows template groups, meaning different image templates generated by the same prompts. These images are supposed to look different. Each image is itself a template; for example, generating “X Area Rug” with different seeds may produce separate images identical everywhere except the rug itself.
>
> **source tracing relies on Google Lens searches and manual inspection. Lack of strict or comprehensive evaluation somewhat undermines the soundness of your findings.**
>
>  While we rely on human evaluation to detect copied material, we reduce subjectivity through a systematic user study which verifies our claims and achieves statistical significance. We do not use similarity metrics because they are not well suited for template-level memorization, where copied content may appear in the background or differ slightly in structure.
>
> **The similarity threshold for CLIP cosine similarity (0.95) is selected empirically without justification or ablation.**
>
> While your comment is correct, this threshold is not the essence of our method. Values in the range 0.9–0.99 behaved as expected, with higher values missing cliques and lower values producing false positives, so 0.95 was a reasonable compromise. We will add a more detailed explanation to the paper.
>
> **Experimental scale is limited. Only 112 collocations are tested, In-The-Wild evaluation uses only 35 candidate prompts.**
>
>  As we report in the results section, our method produced 11,400 images and identified 64 templates, comparable to Somepalli et al. We focused on total generated images and identified templates rather than collocations for consistency with prior work. Fig. 4 suggests that increasing collocations would increase the number of identified templates at a similar rate.
> ## Key Questions:
>
> **What does the proposed attack perform under some mitigation methods like Di et al. (2025a)?**
>
> This is a good suggestion. Our attack shows success on modern models such as SD3.5, suggesting resilience to newer mitigation methods. We conducted preliminary tests with Chen (2025), where the method detected memorized prompts but failed to mitigate in many cases, either still reconstructing TMI or degrading image quality and prompt coherence. We plan to include such evaluation in the paper.
>
> **Do such template-memorized images exist and be able to be discovered by your method for modern commercial models like GPT, Grok, Gemini, etc.?**
>
>  In subsection 4.2 we show template-memorized images in modern text-to-image models such as Flux and SD 3.5. However, your examples are LLMs, so we are not sure what template-memorized images would mean in that context, feel free to clarify.
>
> **How many reconstructed images were verified against training datasets (e.g., LAION) rather than external web search?**
>
>  We did not directly verify against LAION, but we recognized some templates from the standard list of memorized prompts from Webster (2023) and examples shown in prior works such as Di et al. (2025a).
>
> **What is the failure rate of the segmentation-based editable region masking, and how does it affect duplicate detection accuracy?**
>
> Out of the 112 collocations tested, 27 did not have a suitable segmentation category. For the rest, the failure rate varies across templates and samples, and detection accuracy is affected by segmentation accuracy. Even when segmentation failed, cliques sometimes still formed because different seeds produced variations of the same template with the same descriptor. We will consider adding a more exact performance assessment.

---

> > ### Author Rebuttal · Reviewer_zj6c · 2026-04-02
> >
> > Thanks for your rebuttal. Modern commercial models (or agents) can also generate images rather than only focus on LLM (text). So are there similar TMIs (e.g., querying with same/similar prompts) when use them to generate images, if they are, how to find them?

---

> > > ### Author Response · Authors · 2026-04-05
> > >
> > > Thanks for the clarification! This is an interesting and important question. Indeed, much of the day to day interaction between text2image models and real-users (in the wild) occurs through an intermediary involving these models. However, the lack of a public API for generating images through these agents and their black-box nature (e.g., lack of access to prompt expansion and potentially unknown other mitigations applied pre-, in-, or post-inference) make a methodological evaluation of TMI generation in these systems challenging. Nonetheless, we agree that attacking these systems end-to-end is interesting, and defer this exploration for future work.
> > >
> > > That said, following your query, we dug a bit deeper. We found that image generation in agents like the ones you mentioned generally follows two key steps: prompt editing by the LLM (such as prompt upsampling for expansion), followed by a call to a commercial T2I model (e.g., Nano Banana in Gemini).
> > >
> > > Therefore, a necessary pre-condition for constructing TMI through the agents you propose is that the T2I models these agents rely upon would produce TMI. We believe this is the case, and qualitative experiments with commercial FLUX models and the prompts we found in our research indeed show that commercial models generate TMI (notably, we led a model to reproduce the beach towel template from Figure 5).
> > >
> > > Still, With this evidence in mind, we still require means to overcome the pre- or post-processing that prompts and generated images may undergo. Our results suggest that exact prompts are not needed for constructing TMI (this is the same reasoning as to how our attack circumvents Chen et al.’s mitigation in our previous reply), and therefore we believe that such end-to-end attacks would be possible.
> > >
> > > We thank you again for your feedback and for engaging with us, and sincerely hope that our response addresses your remaining concerns.

---

### Official Review · Reviewer_z5kg · 2026-03-11

**Soundness:** 2
**Presentation:** 2
**Significance:** 2
**Originality:** 2
**Overall Recommendation:** 4
**Confidence:** 4

**Summary:**

The paper presents a low-resource reconstruction attack targeting template-memorized images in diffusion models. Unlike prior work that relies on training-set prompts, the paper proposes an attack where prompts are constructed from real-world data without direct training data access. Their work claims to be the first to demonstrate that memorized visual content can be reconstructed through seemingly benign prompts that naïve users might naturally generate during normal model usage. Beyond reconstruction, the authors observe previously undocumented phenomena in template memorization, including interpolation, perturbations, and template leakage.

**Compliance With Llm Reviewing Policy:**

Affirmed.

**Final Justification:**

My concerns have been appropriately addressed.

**Key Questions For Authors:**

See the weaknesses section.

Additional Questions:
Q1: From what I understand, this phenomenon of template memorization comes down to near-duplicates in the training set. However, I think this comes down to a problem during model training that near-duplicates are not filtered. Why is it now surprising that the models learn to interpolate and reproduce these template images?

**Limitations:**

Yes

**Strengths And Weaknesses:**

Strengths:
- The paper observes new behavior for template memorized samples such as template leakage. However, as far as I can see, these derived phenomenons are based on qualitative observations.

Weaknesses:
- I don't see a compelling reason to collect images from these websites with the intention that these images are also in the training data. Since the LAION5B dataset is used as a reference, and this dataset is open, why not simply use the open-source dataset?
- I didn't understand what part of the proposed approach would be an attack. While the method does not directly access the training data, the same data is basically just scraped from the web. However, I don't see where this would be an "attack".
- While it is interesting to see the phenomenon of template leakage or template perturbations, I am not sure whether these observations are general enough. Apart from qualitative examples, are there any quantitative measurements or experiments that support these phenomena?


Misc:
- Lin 97: Parantheses for the citation is missing in right column

---

> ### Author Rebuttal · Authors · 2026-03-30
>
> We thank the reviewer for their thoughtful comments and feedback. We believe there may be some misunderstanding regarding our attack, particularly its scope and objective, and we would like to clarify these points. We hope that, with this clarification, the reviewer may reconsider their evaluation. We will also revise the manuscript to better explain the attack’s purpose and context. Please let us know if this addresses your concerns. If we have misunderstood any of the issues raised, we would be glad to provide further clarifications.
>
> Our attack, like previous attacks we compare with, has two phases: First, we design prompts that may generate images from the training set and collect generated images. In the second phase, we find source images on the website (and other works in the training set (LAION5B)) that we can identify as copied.
> Importantly, the first phase is the primary phase. The second phase is secondary and mainly serves to validate that the attack was successful and that the generated images were indeed copied. Whether the source images are found on the website or in LAION5B is inconsequential from our point of view.
> When we claim that our attack is “low resource”, we mainly refer to the first phase. The main challenge is designing the prompts, and here it matters whether the attack involves access to the training set and whether the prompts are natural, as we elaborate in the paper.
> ## Weaknesses:
> **I don't see a compelling reason to collect images from these websites with the intention that these images are also in the training data. Since the LAION5B dataset is used as a reference, and this dataset is open, why not simply use the open-source dataset?**
>
> Images we collected from websites were used only to validate that the attack succeeded. As clarified above, it is inconsequential whether we collected these images from websites or from LAION5B. Both would demonstrate that the attack was successful and the images were copied.
> Importantly, previous attacks also engage with LAION5B in the first phase when they search for candidate images and prompts and this is where we differ. As we discuss in the paper, this results in more generic prompts, fewer generated images than Carlini with a comparable number of identified images, and no reliance on highly specific prompts that could be mitigated.
>
> **I didn't understand what part of the proposed approach would be an attack. While the method does not directly access the training data, the same data is basically just scraped from the web. However, I don't see where this would be an "attack".**
>
> As discussed, the objective of the attack, ours and previous ones, is to generate images that are copies of training data. The goal was not to find copied images, but to generate them through prompts. We will reframe and add more context in the manuscript for clarification.
>
> **While it is interesting to see the phenomenon of template leakage or template perturbations, I am not sure whether these observations are general enough. Apart from qualitative examples, are there any quantitative measurements or experiments that support these phenomena?**
>
> We support these phenomena by a user study, as well as a synthetic experiment intended to induce such phenomena by fine tuning the model, which is reported in the appendix. As discussed in the paper, similarity metrics often miss this form of reconstruction.
>
> Thank you for identifying typos, we will fix those!
>
> ## Additional Questions:
>
> **Q1: From what I understand, this phenomenon of template memorization comes down to near-duplicates in the training set. However, I think this comes down to a problem during model training that near-duplicates are not filtered. Why is it now surprising that the models learn to interpolate and reproduce these template images?**
>
> We agree that near-duplicates in the training data are likely one of the causes of template memorization, and prior work has shown diffusion models can memorize training samples. Nevertheless, several aspects are still surprising.
> First, we show that copied images may be induced by natural prompts and not necessarily require the original training captions.
> Second, we observe that memorized and generalized regions can co-exist within the same generated image, where a fixed region is reproduced consistently while other regions remain editable.
> Third, interpolation between seeds associated with different templates produces hybrid images combining elements from multiple templates rather than collapsing to a single memorized example.
> Importantly, these phenomena - template-bounded editability, interpolation between memorized templates, and template perturbations - have not been previously studied in memorization attacks, mitigation methods, or common memorization metrics.

---

> > ### Author Rebuttal · Reviewer_z5kg · 2026-04-01
> >
> > Thank you for the clarification. I agree that the main contribution is in the generation phase, not in the post hoc validation of copied images.
> >
> > However, my core concern remains unresolved: although the method may not use the exact training samples or captions, it still appears to rely on substantial prior knowledge about the training distribution in order to construct effective prompts. In fact, the paper explicitly states: “Instead, we compile a list of candidate expressions associated with e-commerce websites known to appear in LAION-5B.”
> >
> > This is precisely the issue I was pointing to. In my view, this means the attack is not truly training-data agnostic, but rather replaces access to exact training instances with access to informative priors about the training corpus. That is a materially weaker and more favorable threat model than the framing currently suggests.
> >
> > In particular, for closed models with unknown or proprietary training data, it is much less clear whether an attacker could construct similarly effective prompt sets without such source-domain knowledge. My concern is therefore not about the validation phase, but about the prompt-construction phase itself, which appears to depend on privileged assumptions about where memorized content comes from.
> >
> > I would encourage the paper to either narrow this claim or evaluate attack performance when these assumptions are weakened or unavailable.

---

> > > ### Author Response · Authors · 2026-04-01
> > >
> > > Thank you very much for engaging and for the followup questions. First, let us clarify that you are absolutely correct. We indeed rely on prior knowledge about the training distribution, and as you write, this is explicitly acknowledged in text. The attack is **not** training-data agnostic but only avoids direct access to the training data and it should be clarified.
> > >
> > > We intend to clarify this in the introduction (third paragraph, line 40, second column) when we introduce our attack. Specifically, we will rephrase and use your wording: Instead of merely saying that we don’t use training data (“.. without relying on training data”), we will write explicitly that **“we replace access to exact training instances with access to informative priors about the training corpus.”** We also intend to clarify this point throughout the text, wherever confusion is possible, so it will not sound as if we claim to be training-data agnostic.
> > >
> > > Finally, we would like to reiterate that the required prior is relatively weak (e.g., assuming that widely occurring web domains are likely to appear in large-scale datasets). Moreover, while we exploit prior knowledge to identify potential prompts, Sec. 4.4 shows prompts that we identified but were written and used by real users without any intentions. Lastly, our attack remains effective over SD3.5 and Flux where indeed the training dataset is not completely disclosed. To the best of our knowledge, we are the only attack where memorized images were extracted from these models. Overall, we will clarify this assumption and its limitations more explicitly in the revision. Thank you very much for this valuable discussion which will help to improve the final text.

---

### Official Review · Reviewer_JFqM · 2026-03-13

**Soundness:** 3
**Presentation:** 2
**Significance:** 3
**Originality:** 3
**Overall Recommendation:** 4
**Confidence:** 3

**Summary:**

This paper reveals that text-to-image models memorize template-memorized images and can unintentionally regenerate them. A low-resource reconstruction attack is designed to obtain template-memorized images, using only benign prompts and requiring little access to the training data. The paper also analyzes the interpolation, perturbation, and template leakage phenomena in template memorization.

**Compliance With Llm Reviewing Policy:**

Affirmed.

**Final Justification:**

This paper uncovers an interesting yet underexplored vulnerability in text-to-image models. Although the quantitative evaluation and mitigation strategies are still somewhat lacking, the authors’ rebuttal addressed my concerns.

**Key Questions For Authors:**

Please address the weaknesses listed above.

**Limitations:**

It discussed the limitation about manual inspection in the paper.

**Strengths And Weaknesses:**

Strengths:
1. The paper exposes security risks of text-to-image models by showing that natural prompts can retrieve template-memorized images.
2. It introduces a low-resource, black-box reconstruction attack that requires no access to the original training data.
3. The discovery of interpolation, perturbation, and template leakage phenomena within template memorization offers valuable insights for future research.

Weaknesses:
1. There is a lack of sufficient quantitative experiments, for example, attack success rates and quality metrics of recovered template-memorized images. As a result, it is not easy to judge the attack’s effectiveness.
2. Although it highlights the risks of template-memorized images, the paper does not propose substantial defense or mitigation strategies.
3. Many figures are used to illustrate the attack’s effects, but their ordering and explanation are unclear, hindering comprehension of the paper.

---

> ### Author Rebuttal · Authors · 2026-03-30
>
> We thank the reviewer for their comments and constructive critique. We hope that the clarifications below will help address the concerns raised and encourage a reconsideration of the score, particularly in light of the paper’s strengths. As the reviewer notes, a key contribution of our work is the identification of novel risks, and we believe it is important for such risks to be brought to the attention of the Machine Learning community.
>
> Regarding the concerns: (1) we agree that the organization of the paper can be improved and will revise it accordingly. However, we believe this issue is primarily one of presentation rather than substance. (2) While we do not propose a mitigation, the primary goal of this work is to identify previously unrecognized risks. We believe that surfacing such vulnerabilities is an essential step, and that requiring immediate mitigation may unintentionally discourage the disclosure and study of important security issues. (3) Concerning the questions around the quantitative evaluation, we provide additional clarification below on our methodology, comparisons, and how these can be more clearly presented in the final version.
>
> Below, we address each of these points in detail.
>
> ## Weaknesses:
> **There is a lack of sufficient quantitative experiments, for example, attack success rates and quality metrics of recovered template-memorized images. As a result, it is not easy to judge the attack’s effectiveness.**
>
> We thank the reviewer for this comment. We would like to clarify that we do report success rates and compare these to previous attacks for judgement. We will highlight this comparison by adding a detailed table (this is slightly delicate as we identify templates where each template consists of more than a single image, nevertheless highlighting the comparison in a table will benefit readability).
>
> The results are currently presented at the second paragraph of the Result section. Overall we generate and report 11,400 images from which 64 copied templates are identified. This is compared with Somepalli et al. that reports generating a similar order of magnitude of images (from which they identified 109 images) and it is compared to Carlini et al. that generated an order of 175 million images (from which they identified 107 images). We neglected to state the number of identified images by previous work, which are all comparable, but this is crucial for correct assessment and we apologize for that. We will add these numbers.
>
> Regarding Quality metrics:
>
> We reported a quantitative user study that demonstrates that template-memorized images are consistently identified and images are recognized in a statistically significant manner. We found that similarity metrics and automated methods are not well-suited for our setting. This is not a limitation of our method, but is a limitation of similarity scores. For example, all of the samples in our user study had a greater distance than 0.15 that Carlini (2023) used to indicate near-identical images, yet more than 70% of the users could identify them as copied.
>
> Prior work focused on verbatim copies, where metrics like SSCD or edge detection were incorporated in the pipeline. In contrast, our work targets template-level memorization, where the reproduced content may be in the background, or differ in minor structure. This made standard similarity metrics unreliable, and we observed that edge consistency frequently failed to flag images that, to a human observer, were clearly copied.
>
> We will add this explanation and discuss the limitations of these automated quality metrics.
>
> **Although it highlights the risks of template-memorized images, the paper does not propose substantial defense or mitigation strategies.**
>
> We indeed did not focus here on mitigation strategies and our focus was mainly in investigating the weakness. We agree that developing a substantial and practical mitigation approach would be a strong and valuable contribution, and we are in fact actively pursuing several ideas and methods ourselves. That being said, the merit of our paper is in exposing the risk, which is not less important. Moreover the difficulty of devising effective defenses should not prevent the disclosure and discussion of these vulnerabilities.
>
> **Many figures are used to illustrate the attack’s effects, but their ordering and explanation are unclear, hindering comprehension of the paper.**
>
> Thanks. In the revision, we will reorganize the figures to better match the narrative flow, and expand the captions and in-text explanations to clarify their role and interpretation.

---

> > ### Author Rebuttal · Reviewer_JFqM · 2026-04-03
> >
> > The authors' responses have addressed my concerns. While the paper offers no mitigation strategies, I agree that the disclosure of TMI vulnerabilities is interesting and important. I also acknowledge that employing automated quality metrics to measure template-level memorization is non-trivial. I will update my score in light of the paper's overall quality and contribution.

---

> > > ### Author Response · Authors · 2026-04-05
> > >
> > > Thank you for acknowledging our rebuttal! We deeply appreciate the positive reception of our rebuttal and the willingness to update the score.

---

### Official Review · Reviewer_iEoB · 2026-03-13

**Soundness:** 3
**Presentation:** 4
**Significance:** 2
**Originality:** 4
**Overall Recommendation:** 4
**Confidence:** 4

**Summary:**

The paper investigates a memorization phenomenon in diffusion models: recurring layouts and visual structures can sometimes be memorized during the training of text-to-image diffusion models. Based on this finding, the paper proposes a template reconstruction attack, in which attackers extract a group of images from diffusion models that share similar structures with some training data.

**Compliance With Llm Reviewing Policy:**

Affirmed.

**Final Justification:**

The authors’ rebuttal has resolved most of my concerns. I believe my questions and requested changes will be addressed in the revision, so I currently support this work.

**Key Questions For Authors:**

1. Reconstruction attacks generally target specific samples rather than a group or pattern of images, so the frame of template reconstruction attack does not fully make sense to me. I do think the paper studies an interesting memorization phenomenon, but the reconstruction attack framing seems overstated.
2. The authors use MaskFormer and SegFormer to divide image regions and estimate editable region masks. Does the model output directly indicate which regions are editable and which are fixed, or is there additional manual processing involved?
3. Besides human evaluation, would it be possible to introduce vision language models (ChatGPT, Claude, etc.) to evaluate whether a generated image is derived from a source image?

**Limitations:**

yes

**Strengths And Weaknesses:**

**Strengths**

- The paper studies an interesting phenomenon: template memorization.
- The assumption of the proposed attack is realistic: black box access, no access to the training data, and benign prompts.
- The experiments are comprehensive and include multiple text-to-image models.
- The analysis of template memorized images is insightful.

**Weaknesses**

- The paper frames the template memorization phenomenon in text-to-image diffusion models as a reconstruction attack. However, reconstruction attacks usually refer to verbatim or near verbatim copies. I would not describe two images with only similar templates as one being reconstructed from the other. This setting seems less severe than standard reconstruction attacks, since it is natural for these models to memorize recurring templates, especially for frequently occurring image patterns in the training data. In this sense, the phenomenon looks more like a form of generation behavior than a reconstruction attack.
- The evaluation heavily relies on human evaluation, which is not automated or standardized enough and may introduce subjective bias.

---

> ### Author Rebuttal · Authors · 2026-03-30
>
> We thank the reviewer for their thoughtful and positive feedback. Below we address the concerns and key questions.
>
> ## Weaknesses
>
> **The paper frames the template memorization phenomenon in text-to-image diffusion models as a reconstruction attack. However, reconstruction attacks usually refer to verbatim or near-verbatim copies. I would not describe two images with only similar templates as one being reconstructed from the other. This setting seems less severe than standard reconstruction attacks, since it is natural for these models to memorize recurring templates, especially for frequently occurring image patterns in the training data. In this sense, the phenomenon looks more like generation behavior than a reconstruction attack.**
>
> To distinguish from standard reconstruction attacks, we can replace the term “reconstruction attack” with “template reconstruction attack.” However, we do not think this setting is less severe. As we discuss in the paper, such copying is more challenging to detect, especially with automated methods, yet still contains verbatim elements taken from the training set. In extreme cases, these elements can even be real human faces, which poses the standard risks involved with reconstruction attacks regarding privacy, security, and copyright.
>
> We also note that this addresses the reviewer’s later question regarding whether reconstruction attacks should target specific samples rather than groups of images. Our attack demonstrates verbatim copying of specific objects or regions, not only abstract patterns. Moreover, previous reconstruction attacks also rely on redundancy and repeated samples in the training set, so the distinction between targeting specific samples and repeated templates is somewhat ambiguous.
>
> **The evaluation heavily relies on human evaluation, which is not automated or standardized enough and may introduce subjective bias.**
>
> While we indeed rely on human evaluation to identify copied material, similarity metrics and automated methods are not well suited for our setting. This is not a limitation of our method, but rather a limitation of similarity scores. For example, all of the samples in our user study had a greater distance than 0.15, which Carlini (2023) used to indicate near-identical images, yet more than 70% of the users could identify them as copied.
>
> To mitigate potential subjectivity, we performed a systematic user study, used appropriate controls, and published the results along with confidence intervals to provide consistent, repeatable, and quantifiable assessments. Overall, we can conclude that humans can reliably detect the copied elements, and this claim is not subjective.
>
> Prior work focused on verbatim copies, where metrics like SSCD or edge detection were incorporated in the pipeline. In contrast, our work targets template-level memorization, where the reproduced content may be in the background or differ in minor structure. This made standard similarity metrics unreliable, and we observed that edge consistency frequently failed to flag images that, to a human observer, were clearly copied.
>
> We will add this discussion to the paper in order to clarify. Thank you.
>
> ## Key Questions for Authors
>
> **The authors use MaskFormer and SegFormer to divide image regions and estimate editable region masks. Does the model output directly indicate which regions are editable and which are fixed, or is there additional manual processing involved?**
>
> There is no additional manual processing involved. As described in Subsection 3.2, we manually selected the segmentation category closest to the collocation, and we binarized the output to the selected category versus the rest.
>
> **Besides human evaluation, would it be possible to introduce vision-language models (ChatGPT, Claude, etc.) to evaluate whether a generated image is derived from a source image?**
>
> Thank you for this constructive suggestion. We performed preliminary experiments on Claude and Gemini through their web interfaces, and they could easily identify if a generated image is derived from a source. We agree that incorporating evaluations from such models is a valuable contribution, and we will report a thorough examination in the paper.
>
> That being said, as with automated metrics, given that humans can reliably identify copied elements, a failure by these models to detect copying would likely reflect limitations of the models themselves rather than the absence of copying.

---

> > ### Author Rebuttal · Reviewer_iEoB · 2026-04-02
> >
> > Thank the authors for the detailed clarification, which addressed most of my concerns.

---

> > > ### Author Response · Authors · 2026-04-05
> > >
> > > Thank you for the acknowledging our rebuttal! We appreciate your feedback and your positive assessment.

---

### Decision · Program_Chairs · 2026-04-30

**Decision:**

Accept (regular)

**Comment:**

The authors have addressed all reviewers’ concerns, and all reviewers recommend acceptance of the paper. Based on the consistency of the reviews and the satisfactory rebuttal, the AC supports an acceptance decision. The authors should include appropriate discussion to address the reviewer critiques in the camera ready.